# SK119, a Novel Shikonin Derivative, Leads to Apoptosis in Melanoma Cell Lines and Exhibits Synergistic Effects with Vemurafenib and Cobimetinib

**DOI:** 10.3390/ijms23105684

**Published:** 2022-05-19

**Authors:** Nadine Kretschmer, Christin Durchschein, Antje Hufner, Beate Rinner, Birgit Lohberger, Rudolf Bauer

**Affiliations:** 1Institute of Pharmaceutical Sciences, Department of Pharmacognosy, University of Graz, Beethovenstr. 8, 8010 Graz, Austria; nadine.kretschmer@uni-graz.at (N.K.); christin.durchschein@hotmail.com (C.D.); rudolf.bauer@uni-graz.at (R.B.); 2Division of Biomedical Research, Medical University Graz, Roseggerweg 48, 8036 Graz, Austria; beate.rinner@medunigraz.at; 3Institute of Pharmaceutical Sciences, Department of Pharmaceutical Chemistry, University of Graz, Universitaetsplatz 1, 8010 Graz, Austria; antje.huefner@uni-graz.at; 4Department of Orthopedics and Trauma, Medical University Graz, Auenbruggerplatz 5, 8036 Graz, Austria

**Keywords:** shikonin dimer, shikonin, melanoma, apoptosis, synergism, vemurafenib

## Abstract

Melanoma is a complex and heterogenous disease, displays the deadliest form of skin cancer, and accounts for approx. 80% of all skin cancer deaths. In this study, we reported on the synthesis and pharmacological effects of a novel shikonin derivative (SK119), which is active in a nano-molar range and exhibits several promising in vitro effects in different human melanoma cells. SK119 was synthesized from shikonin as part of our search for novel, promising shikonin derivatives. It was screened against a panel of melanoma and non-tumorigenic cell lines using XTT viability assays. Moreover, we studied its pharmacological effects using apoptosis and Western blot experiments. Finally, it was combined with current clinically used melanoma therapeutics. SK119 exhibited IC_50_ values in a nano-molar range, induced apoptosis and led to a dose-dependent increase in the expression and protein phosphorylation of HSP27 and HSP90 in WM9 and MUG-Mel 2 cells. Combinatorial treatment, which is highly recommended in melanoma, revealed the synergistic effects of SK119 with vemurafenib and cobimetinib. SK119 treatment changed the expression levels of apoptosis genes and death receptor expression and exhibited synergistic effects with vemurafenib and cobimetinib in human melanoma cells. Further research indicates a promising potential in melanoma therapy.

## 1. Introduction

According to the World Health Organization (WHO, Geneva, Switzerland), the incidence of skin cancer has been increasing over recent decades, with a global occurrence of 2–3 million cases each year. This means that one in every three cancers diagnosed is a skin cancer. Melanoma accounts for approximately 5% of all skin cancer cases, but displays the deadliest form of skin cancer and accounts for approx. 80% of all skin cancer deaths. Moreover, its incidence has increased faster than the incidence of any other type of cancer [1,2]. Melanoma is a complex and heterogenous disease, making its diagnosis and treatment still extremely difficult today. Even if improvements in melanoma therapy were achieved with, for example, BRAF and PD1 inhibitors, the treatment of metastatic melanoma remains challenging [3]. Therefore, there is still an urgent need for the discovery and development of novel potential melanoma drugs.

Natural products have always played a crucial role in the discovery of novel anti-cancer drugs, and traditional remedies still dominate therapeutic measures all around the world [4,5]. Today, almost 80% of all approved, small-molecule anti-cancer drugs are natural products per se, or derived from them [4]. A famous example is camptothecin, a monoterpenoid indole alkaloid isolated for the first time in 1966 from the Chinese tree *Camptotheca acuminata* Decne (Nyssaceae) [6], which mediates its anticancer effects by inhibiting DNA topoisomerase I. The synthetic analogs of camptothecin, such as topotecan and irinotecan, were developed to improve clinical efficiency and are in clinical use for more than 25 years, reflecting their important role [7,8]. Moreover, nature continues to be an inspiring source of novel biologically active compounds, however, most isolated natural products will not directly develop into clinically effective drugs. Instead, they serve as a pharmacophore, giving model compounds to develop more effective drugs [5].

In our survey for discovering novel promising anticancer drugs, the constituents of the roots of *Onosma paniculatum* Bur and Franch (Boraginaceae) arose as such candidates [9,10,11,12]. These roots are traditionally used in Chinese medicine to treat different skin disorders in particular, such as measles, skin infections, cuts, and burns and are applied internally and externally [13]. The main pharmacological active constituents are naphthoquinones, such as shikonin and the derivatives thereof. We could show that the most active and promising natural derivative was *β*,*β*-dimethylacrylshikonin (DMAS), which was especially active against a panel of melanoma cell lines [10,11,12]. In ongoing studies, our goal was to modify the structure of DMAS to further optimize its effects. Firstly, among other derivatives, we developed a novel shikonin derivative—cyclopropylacetylshikonin—which was more effective against two metastatic melanoma cell lines than DMAS [14]. A similar derivative bearing another cyclopropane moiety was also shown by our group to exhibit promising effects in melanoma cell lines [15]. In the current study, we report on the synthesis and pharmacological effects of another novel shikonin derivative (SK119), which is active in a nano-molar range and exhibited several promising in vitro effects.

## 2. Results and Discussion

### 2.1. Synthesis and Structure Elucidation of SK119 

In our studies about the structure–activity relationship of DMAS and its derivatives, we analyzed the influence of the configuration of the side chain. Initially, we aimed to prepare dimethyl-acryl-alkannin from shikonin and dimethyl acrylic acid, via the Mitsunobu reaction, which is similar to a procedure that was used by Liao et al. for the acylation of a α hydroxy phenyl compound [16]. However, DEAD (diethyl azodicarboxylate) was replaced by DIAD (diisopropyl azodicarboxylate), which is more convenient to handle. Though the R_f_-value of the reaction product (TLC on silica) was close to DMAS, the spot was more purple in color. After isolation and structure elucidation, the structure of this product was finally determined as the dimer SK119, which is depicted in Figure 1. 

The ^1^H and ^13^C NMR spectra of SK119 in CDCl_3_ (Figure 2A,B, NMR data: Table 1) reveal a dimeric structure of SK119. Hence, the ^13^C NMR spectrum showed the nearly 30 signals, including four carbonyl signals (δ = 182 to 187 ppm). The ^1^H NMR spectrum (Figure 2A) shows two shikonin-like moieties, A and B, as exemplified by the four singlets of phenolic OH protons (δ = 12.4 to 12.7 ppm) and the four singlets of the four methyl groups (δ = 1.6 to 1.9 ppm), but only one methylene group (multiplet at δ = 2.8 ppm). Two doublets at δ = 6.04 and 6.58 ppm resp. and a doublet of doublets at δ = 7.41 ppm with coupling constants of 15.6 and 11.3 Hz indicate the elimination of water in one side chain, resulting in an (*E*)-4-methylpenta-1,3-dienyl group in the shikonin-like moiety A. There is still a 4-methylpent-3-enyl group attached to the shikonin-like moiety B, but the α-hydroxy group is replaced by a less polar substituent. DEPT-HSQC and ^1^H,^1^H COSY (see Appendix A) show the signals of the α-CH (CH-11′) at δ = 4.67 ppm and δ = 38.8 ppm. DEPT-HSQC and HMBC (see Appendix A) allowed to assign all of the hydrogen and carbon atoms of subunit B, which still bears a hydrogen in position 3. Overlay and very similar chemical shifts of several signals prevent the distinct assignment of all signals of entity A, but couplings of α-CH of the 4-methylpent-3-enyl chain (CH-11′) with carbonyls of both quinones confirm this CH group as a linker of both quinones. In HMBC, the carbon signal at 143.8 ppm, which belongs to naphthazarine A, couples with α-protons of both of the side chains, whereas the two α-protons couple to different carbonyls of moiety A. This, as well as the NOEs (see Appendix A) between the two α-CH of the side chains (H-11 and H-11′), prove the proximity of both chains. The C-2 of moiety A is assumed to show cross peaks in HMBC with α and β protons of the 4-methylpenta-1,3-dienyl group, but probably due to overlay with signals of C-3 or δ-C (C-14), the signal of C-2 could not be identified. Therefore, SK119 was also measured in benzene-d_6_ (see Appendix A). As depicted in Figure 3, C-2 appears as an additional signal at 144.1 ppm close to C-3 (144.7 ppm) coupling with the α protons of both side chains.

This type of dimeric shikonin is reported for the first time. Among the known dimeric shikonin derivatives (for examples, see Appendix A, chapter 3), shikometabolin C is the only one with direct connection of α-C of the side chain with the quinoid ring of another shikonin unit. However, it shows an additional ring closure resulting in a pentacyclus with the loss of one quinonoid pattern [17,18]. Shikometabolin C and other types of known shikonin dimers are presented in the Appendix A. 

### 2.2. Screening of SK119 against Different Melanoma Cell Lines

SK119 was firstly screened against a panel of melanoma cell lines. To investigate the effects of SK119 on non-tumorigenic cells, fresh, isolated human adult fibroblasts (FS-1), and embryonic kidney cell line HEK-293 were used. To cover a broad range of different melanoma types, we used the Sbcl2 cell line as an example of an early state of tumor progression (cutaneous melanoma, radial growth phase) and three melanoma cell lines derived from melanoma metastases (WM9, WM164, and MUG-Mel2), because melanoma metastases often respond poorly to cancer therapeutics. WM9 cells originated at the left axillary node and are BRAF mutated. WM164 cells were obtained from the right upper arm and represent a stage IV superficial spreading melanoma and are also BRAF-mutated. BRAF mutations are the most frequent mutations in melanoma, occurring in about 68% of melanoma metastases, 80% of primary melanoma, and 82% of nevi [19]. It is assumed that these mutations are a critical step for the initiation of melanomagenesis. The most frequent mutation is characterized by a valine-to-glutamic acid substitution at codon 600 (V600E). This leads to a ten-fold higher kinase activity, which results in increased cell proliferation and a lower incidence of apoptosis [19,20]. Currently, melanoma carrying a BRAF^V600E^ mutation are typically treated with a combination of BRAF and MEK inhibitors. However, tumor resistances and recurrences often occur [21]. In addition, we used a cell line which carried no BRAF, but a NRAS mutation. This MUG-Mel2 cell line was isolated and established from a very fast-growing metastasis at the left shoulder [22]. NRAS mutations occur in about 25% of all melanomas. In the case of a NRAS mutation, no targeted inhibitors are in clinical use so far. NRAS mutated melanomas are mostly only treated with MEK inhibitors, leading to an even lower therapeutic success rate [23].

When compared to the most active, isolated derivative DMAS [10], SK119 was on average four- to five-fold more active, as shown by their IC_50_ values (Table 2). In the case of MUG-Mel2 cells, SK119 exhibited even a 10-fold lower IC_50_ than DMAS, indicating a 10-fold higher effect.

In the literature, dimeric shikonin derivatives described so far are vaforhizin and iso-vaforhizin and several shikometabolines, such as shikometabolines A–H [17,18,24,25,26,27]. However, these dimers are quite heterogenous in structure and, therefore, difficult to compare. Shikometabolines were, for example, found after incubation of shikonin with human intestinal bacteria [17,18]. In a study of Min et al. (2000), shikometabolines A–D were tested for their cytotoxicity against five cancer cell lines (prostate, lung, kidney, colon, and leukemia) [28]. However, all four derivatives were less cytotoxic than SK119. In another study, shikometabolin H was shown to be cytotoxic with a relatively high IC_50_ over 30 µM and inhibited the growth of colon cancer cells by affecting STAT3 [27]. Other pharmacological activities besides cytotoxicity were shown, for example, for vaforhizin and iso-vaforhizin, which can be found in different species of the Boraginaceae family, and which exhibited anti-leishmanial activity [24]. Moreover, two acylated derivatives of shikometabolin B exhibited neuraminidase inhibitory activity [25]. 

In summary, however, dimeric shikonins are, as yet, scarcely investigated pharmacologically. In our study, SK119 was also tested in two non-tumorigenic cell lines. FS-1 cells are adult fibroblasts freshly isolated from an apron of fat. HEK-293 cells are a well-known and often used cell line, derived from human embryonic kidneys. Both cell lines were also affected by SK119. For the FS-1 cells, the IC_50_ value was in the same range as for the cancer cells. In the case of the HEK-293 cells, the IC_50_ value was up to seven-fold higher than the IC_50_ value found for the melanoma cell lines. The cytotoxicity of chemotherapeutics towards healthy cells is a known problem in cancer therapy, and can lead to undesired side effects. Vinblastine, for example, is an alkaloid found in *Vinca rosea* and in clinical use for many years for the treatment of e.g., breast cancer, Hodgkin’s disease, non-Hodgkin’s lymphoma, and Histiocytosis X. When tested in vitro, it is active in the same concentration range towards melanoma cells and non-tumorigenic lung fibroblasts [10]. Nevertheless, further studies will be necessary to investigate the risk–benefit ratio of our novel compound.

The next step was to investigate whether SK119 leads to apoptosis induction, as is reported for several other shikonin derivatives, including DMAS [12,29]. Therefore, a Caspase-Glo^®^ 3/7 assay was performed, which measures the activity of caspases 3 and 7 belonging to effector caspases and, therefore, displaying key enzymes during the process of apoptosis. For comparison reasons, the cell lines WM9 and MUG-Mel2 were used for the ongoing studies. Staurosporine, a protein kinase inhibitor, was used as the positive control and showed a rapid and strong increase in caspase 3/7 activity. This compound has been characterized as a strong inducer of apoptosis in many different cell types [30]. As shown in Figure 4A, SK119 led to a strong increase in caspase 3/7 activity, especially after 24 h, which is indicative of apoptosis induction. Based on these data, we investigated the apoptotic induction in detail.

### 2.3. SK119 Treatment Changed Expression Levels of Apoptosis Genes and Death Receptor Expression

Firstly, we investigated the internal apoptosis pathway by analyzing the effect of SK119 on the pro-apoptotic gene BAK, the anti-apoptotic gene Bcl-2, and the antagonist NOXA. Melanoma cell lines were treated with 0.75, 1.5, and 3 µM SK119, followed by total protein extraction after 24 h incubation and expression levels were determination using Western blot analysis. The pro-apoptotic gene BAK and the antagonist NOXA were upregulated in both cell lines in a dose-dependent manner, whereas the anti-apoptotic gene Bcl-2 reduced accordingly (Figure 4). This apoptotic induction via NOXA downregulation could already be detected with DMAS in melanoma cells [12].

In the external apoptosis pathway, the tumor necrosis factor (TNF) receptor family, which includes TNF-R1, TNF-R2, Fas, and the TRAIL receptors, plays an important role in the regulation of apoptosis in various physiological systems [31]. Death receptors, as a subgroup of the TNF-receptor superfamily, are characterized by a highly conserved extracellular region containing cysteine-rich repeats and a conserved intracellular region of about 80 amino acids termed the death domain. This domain is required for the transmission of the cytotoxic signal by recruiting adaptor proteins (FADD, TRADD, RIP) resulting in the activation of caspases [32]. TNF-R1 and TNF-R2, the two receptors for TNF-α, can mediate distinct cellular responses [33]. Total protein analysis 24 h after SK119 treatment revealed a decrease of TNF-R1 and TRAIL-R2 protein expression in WM9 melanoma cells, whereas the TNF-R2 expression increased in both cell lines (Figure 5). The expression of the other two receptors was only very weak in MUG-Mel2. The corresponding adaptor protein TRADD increased, and FADD and RIP diminished in our cellular system. As SK119 is a completely new derivative, no comparable results could be found in the literature.

Heat shock proteins (HSPs) are a set of highly conserved proteins and powerful chaperones whose expression is induced in response to a wide variety of physiological and environmental insults, including anticancer chemotherapy, thus allowing the cell to survive lethal conditions [34]. Secondly, HSPs are powerful anti-apoptotic proteins, associating with key effectors of the apoptotic machinery, and thereby interfering with this cell death process at different stages. At the post-mitochondrial level, it was shown that HSP27 binds to cytochrome c [35], and HSP70 binds to the apoptotic protease activating factor 1 (Apaf-1), thereby inhibiting caspases’ activation and apoptotic cell death [36]. SK119 treatment caused a dose-dependent increase in the expression and protein phosphorylation of HSP27 and HSP90 in WM9 and MUG-Mel 2 cells (Figure 5). 

### 2.4. SK119 Exhibited Synergistic Effects with Vemurafenib and Cobimetinib

Finally, we investigated the effects of SK119 in combination with vemurafenib and cobimetinib. One of the central pathways that is most dysregulated in melanoma is the RAS-RAF-MEK-ERK-MAP kinase pathway. In most cases, a BRAF mutation is found [37]. Mutated BRAF proteins have an increased kinase activity, leading to uncontrolled cell growth and cancer development [38]. The second most common key driver mutation can be found in NRAS, which also plays a key role in this pathway. A few years ago, BRAF inhibitors, such as vemurafenib and dabrafenib, revolutionized melanoma treatment. However, adverse side effects and especially the development of resistance within the melanoma are common, which leads to a poor prognosis [39]. In the case of the NRAS-mutant melanomas, tumors exhibit an aggressive clinical behavior, are hard to treat, and are associated with a very poor prognosis [40]. These tumors are typically treated with MEK-inhibitors such as cobimetinib. Since the WM9 cells are BRAF mutated and the MUG-Mel2 cells are NRAS mutated, we decided to use vemurafenib for the WM9 cells and cobimetinib for the MUG-Mel2 cells. First of all, we determined the IC_50_ values for both compounds in each cell line using the XTT viability assay (Figure 6A,D). In WM9 cells, we found an IC_50_ of vemurafenib of 82.1 nM. Cobimetinib exhibited an IC_50_ value of 55.2 nM in MUG-Mel2 cells. To study the synergistic or antagonistic effects, cells were treated with different concentrations of SK119 and vemurafenib or cobimetinib, and the results were analyzed using the XTT assay and the free software tool, Combenefit (Cancer Research UK, University of Cambridge, Cambridge, UK) [41]. As shown in Figure 6B,C, we found antagonistic effects in the WM9 cells when SK119 was applied in a very low concentration of 0.3 µM. However, when both compounds were applied at IC_50_, we found a strong synergistic effect. In the case of the MUG-Mel2 cells, the application of both compounds at the approx. IC_50_ also led to synergistic effects. In addition, when SK119 was applied at higher concentrations, it led to antagonistic effects. This also shows that, in the case of combinatorial treatment, the concentration of each compound is of great importance to lead to desirable effects. In general, combinatorial therapy in melanoma is highly recommended to reduce the risk of resistance of the cancer cells and has been shown to dramatically improve response rates, progression-free survival, and overall survival [42]. Therefore, this approach is of high interest and needs to be investigated in more detail in further studies.

## 3. Materials and Methods

### 3.1. Synthesis of 1,4-Dihydro-[2-(1-(1,4-dihydro-5,8-dihydroxy-1,4-dioxonaphthalen-3-yl)-4-methylpent-3-enyl)]-5,8-dihydroxy-3-((E)-4-methylpenta-1,3-dienyl)naphthalene-1,4-dione (SK119)

Shikonin was purchased from Chengdu Biopurify Phytochemicals Ltd. (Chengdu, China). Reagents and solvents were acquired from commercial suppliers. Solvents were dried and purified using standard techniques. Under argon atmosphere, a solution of shikonin (29 mg, 0.1 mmol) and dimethyl acrylic acid (20 mg, 0.2 mmol) in abs. THF (0.83 mL) was cooled to 0 °C. PPh_3_ (105 mg, 0.4 mmol) was added and the solution was stirred for 15 min, followed by addition of DIAD (0.13 mL, 0.75 mmol) in 0.22 mL THF. After stirring for 30 min at 0 °C, cyclohexane was added (1 mL) and the mixture was concentrated in vacuo. The residue was filtered over silica and celite^®^ (2 mm layer each), the products containing fractions were combined, concentrated in vacuo, and submitted to flash chromatography (silica, cyclohexane cyclohexane/CH_2_Cl_2_ = 1:0 to 1:2). PTLC of the raw product (silica, developed three times with cyclohexane/CH_2_Cl_2_ = 2:1 and twice with cyclohexane /CH_2_Cl_2_ = 1:1) resulted in 6.5 mg (24%) SK119.

### 3.2. Spectroscopic Data of SK119 

*Nuclear magnetic resonance* (NMR) spectroscopy data were recorded on a Varian 400 MHz spectrometer (400 and 100 MHz, respectively, Varian, Palo Alto, CA, USA). Chemical shifts are referenced to solvent (7.26 and 77.00 ppm for CDCl_3_ and 7.16 and 128.36 ppm for benzene-d6). The data are depicted in the Results and Discussion sections. Additional data: IR (ATR): 2970 (w); ~2950 (br) (OH); 2911 (w); 2856 (m); 1722 (w); 1640 (s) (C=O); 1568 (s); 1450 (s); 1404 (m); 1262 (s); 1190 (s); 1137 (m); 780 (m); 733 (s) cm^−1^ obtained on a Bruker ALPHA Platinum ATR A220/D-OX (Bruker, Billerica, MA, USA); MS (ESI^-^) *m*/*z* (%): 540.17, calculated for C_32_H_28_O_8_: 540.1784 (Dionex Ultimate 3000 UHPLC, Thermo, San José, CA, USA).

### 3.3. Cell Culture 

Sbcl2, WM164, WM9, and MUG-Mel2 melanoma cells were cultured in RPMI 1640 medium (Gibco^®^, Thermo Fisher Scientific Inc., Waltham, MA, USA), supplemented with 10% fetal bovine serum (FBS) and 1% penicillin/streptomycin solution (both Gibco^®^). Human adult fibroblasts (FS-1) were kindly provided by Prof. Dr. Beate Rinner (Medical University of Graz, Graz, Austria) and HEK293 cultured in Dulbecco’s Modified Eagle Medium (DMEM, Gibco^®^), 2 mM L-glutamine, and 10% FBS. All cells were kept in a humidified 5% CO_2_ atmosphere at 37 °C and passaged at 90% confluence using a trypsin-EDTA 0.25% solution. The cell line was authenticated by STR profiling within the last three years. All experiments were performed with mycoplasma-free cells.

### 3.4. Preparation of Test Compounds 

Test compounds were dissolved in ethanol and freshly diluted with phosphate buffered saline (PBS) before each experiment. The final ethanol concentration was max. 0.5% and did not affect the cells’ behavior, as shown by the control experiments.

### 3.5. Cell Viability Assay

This assay was used for the determination of the IC_50_ values and for the analysis of synergistic and antagonistic effects. For the determination of cell viability, the cell proliferation kit II (XTT) (Roche Diagnostics, Mannheim, Germany) was used, in accordance with the manufacturer’s instructions. In detail, 5 × 10^3^ cells/well were seeded in clear 96-well flat bottom plates and grown for 24 h to allow the cells to adhere. Afterwards, SK119 was freshly diluted with PBS and added to the wells in at least five different concentrations. Control cells were treated with 0.5% ethanol. After a 72 h incubation period, a fresh XTT solution was prepared by mixing 5 mL of XTT with 100 µL of the electron coupling reagent. To each well, 50 µL of this XTT solution was added and cells were incubated at 37 °C for another 2 h. Finally, absorbance at 490 nm with a reference wavelength of 650 nm was measured using a Hidex Sense Microplate Reader (Hidex, Turku, Finland). Cell viability was calculated as a percentage of control cells using the following formula: (absorbance 490 nm–650 nm of treated cells/absorbance 490 nm–650 nm of control cells) * 100. The IC_50_ values were determined using the four parameter logistic curve and SigmaPlot 14.0 (Systat Software, Inc., Düsseldorf, Germany). To analyze the results of the combinatorial treatment, the free available software tool Combenefit (Cancer Research UK, University of Cambridge, Cambridge, UK) was used in accordance with the provider’s recommendations and instructions. Vinblastine at 0.01 µM served as a positive control. 

### 3.6. Caspase-Glo^®^ 3/7 Assay

The Caspase-Glo^®^ 3/7 Assay was purchased from Promega (Madison, WI, USA) and used in accordance with the manufacturer’s instructions. In brief, 100 µL of a 100,000 cells/mL solution was pipetted in white 96-well plates and incubated for 24 h to allow the cells to adhere. Afterwards, SK119 was freshly diluted, added to the cells, and the cells were treated for 6 h, 24 h, and 48 h. Staurosporine (Abcam, Cambridge, UK) at 10 µM served as the positive control. For analysis, a Caspase-Glo^®^ 3/7 reagent was prepared and added to the cells. Finally, luminescence was measured using a Hidex Sense Microplate Reader (Hidex, Mainz, Germany). The assay was performed two times, with three replicates each.

### 3.7. Western Blot Analysis

Whole cell protein extracts were prepared with lysis buffer (50 mM Tris-HCl pH 7.4, 150 mM NaCl, 1 mM NaF, 1 mM EDTA, 1% NP-40, 1 mM Na_3_VO_4_) and protease inhibitor cocktail (P8340; Sigma Aldrich), subjected to SDS-PAGE and blotted onto Amersham™ Protran™ Premium 0.45 µM nitrocellulose membrane (GE Healthcare Life Science, Little Chalfont, UK). Protein concentration was determined with the Pierce BCA Protein Assay Kit (Thermo Fisher Scientific), according to the manufacturer’s protocol. Primary antibodies against Bcl-2, BAK, NOXA, survivin, HSP70, HSP27, phospho HSP27, and the Death Receptor Antibody Sampler Kit, including primary antibodies against Fas, DcR2, DcR3, TRAILR2, FADD, RIP, TNF-R1, TNF-R2, and TRADD (Cell Signaling Technology, Danvers, MA, USA), were used. The β-Actin was purchased from Santa Cruz (Santa Cruz Biotechnology, Santa Cruz, CA, USA). Blots were developed using a horseradish peroxidase-conjugated secondary antibody (Dako, Jena, Germany) at room temperature for 1 h and the Amersham™ ECL™ prime Western blotting detection reagent (GE Healthcare), in accordance with the manufacturer‘s protocol. Chemiluminescence signals were detected with the ChemiDocTouch Imaging System (BioRad Laboratories Inc., Hercules, CA, USA) and images were processed with the ImageLab 5.2 Software (BioRad Laboratories Inc.). Signal intensities were analyzed densitometrically and expressed relative to β-actin, and the values were expressed relative to the control (mean ± SD; *n* = 3).

## Figures and Tables

**Figure 1 ijms-23-05684-f001:**
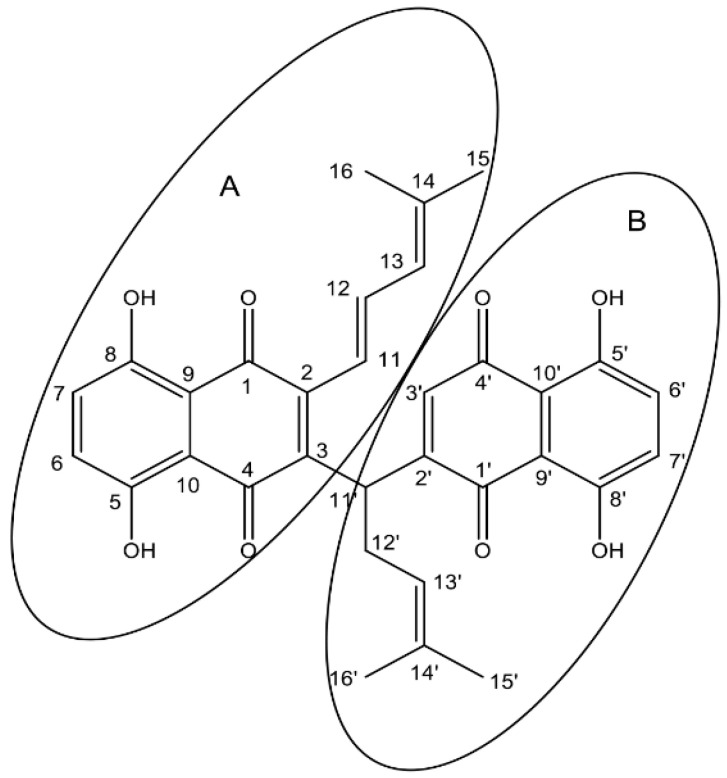
Structure of 1,4-dihydro-[2-(1-(1,4-dihydro-5,8-dihydroxy-1,4-dioxonaphthalen-3-yl)-4-methylpent-3-enyl)]-5,8-dihydroxy-3-((*E*)-4-methylpenta-1,3-dienyl)naphthalene-1,4-dione (SK119). (**A**,**B**) indicate the shikonin-like moieties.

**Figure 2 ijms-23-05684-f002:**
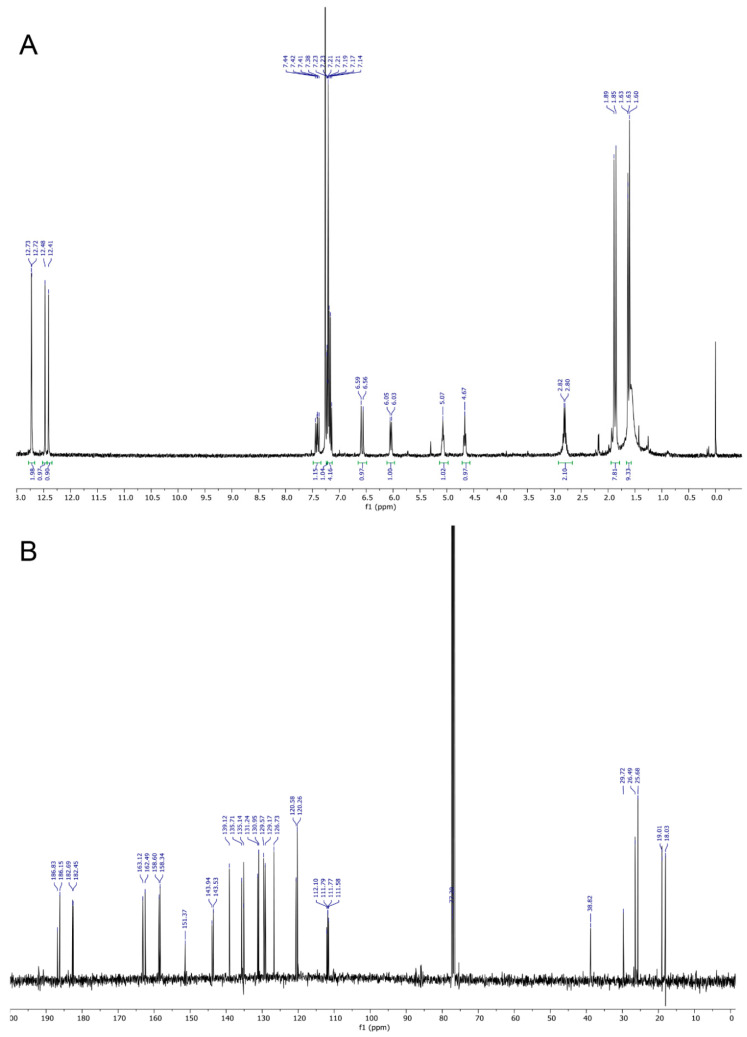
(**A**) ^1^H NMR spectrum and (**B**) ^13^C NMR spectrum of SK119, measured in CDCl_3_.

**Figure 3 ijms-23-05684-f003:**
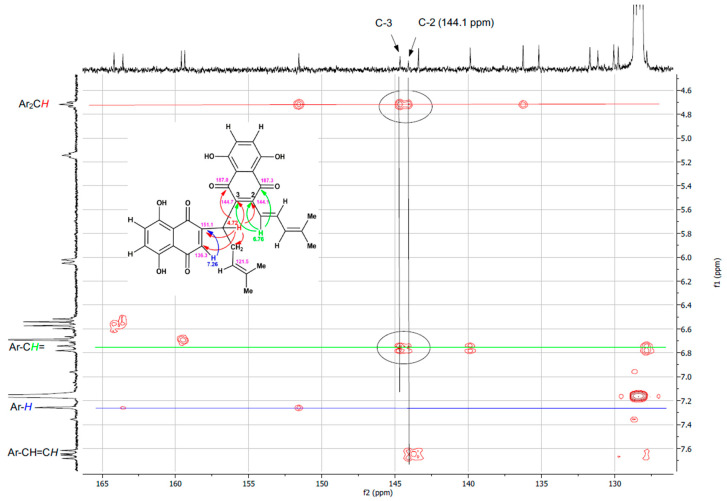
Detail of the HMBC spectrum of SK119, measured in benzene-d6.

**Figure 4 ijms-23-05684-f004:**
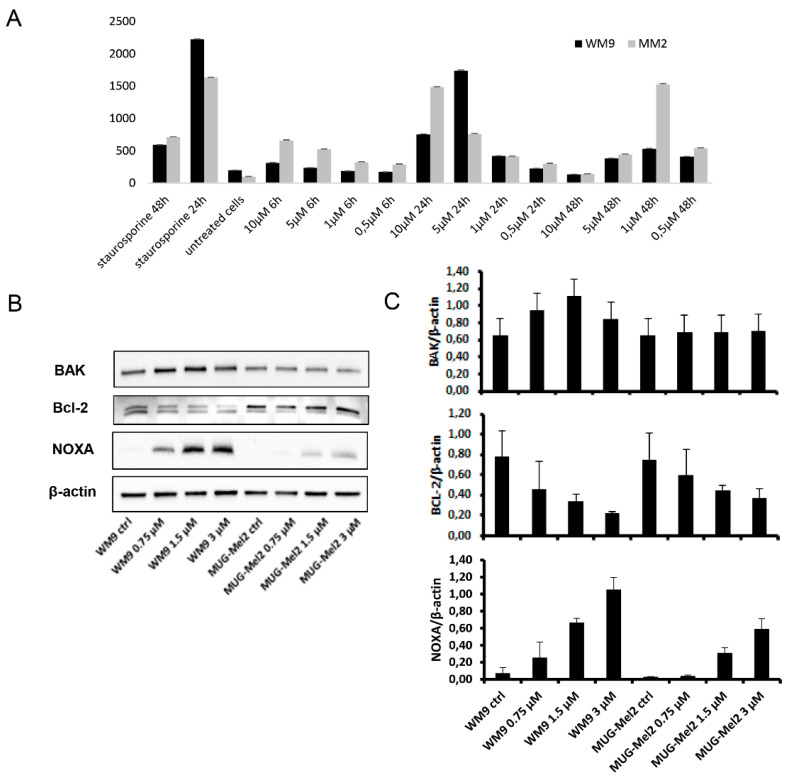
Apoptotic induction by treatment with SK119. (**A**) Results of the Caspase-Glo^®^ 3/7Assay. Treatment of WM9 and MUG-Mel2 cells led to a strong activity increase in caspases 3/7, indicative of apoptosis induction 24 h after treatment (mean ± SD, *n* = 6). Staurosporine (25 µM) served as positive control; (**B**) Western blot analyses were used to verify the expression of the pro-apoptotic BAK, the anti-apoptotic Bcl-2 and the antagonist NOXA at the protein level. One representative blot out of three is shown and β-actin was used as loading control; (**C**) Densiometric quantification are presented (mean ± SD; *n* = 3). All full-length blots are presented in the Appendix A.

**Figure 5 ijms-23-05684-f005:**
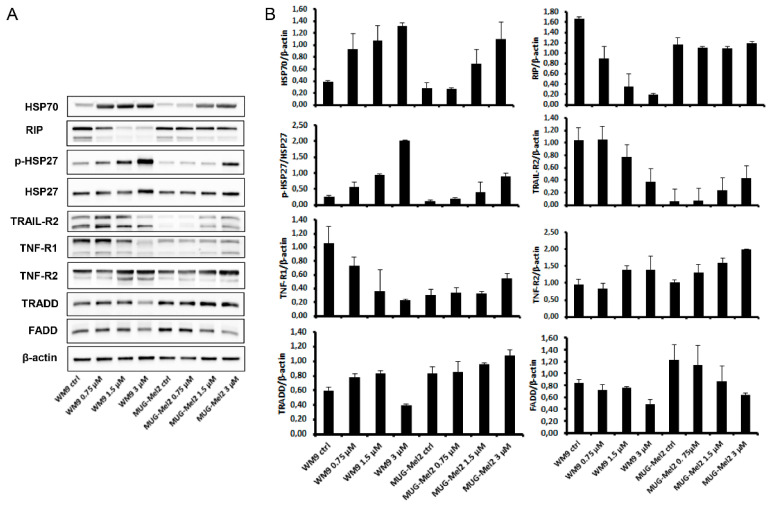
Influence of SK119 treatment on protein expression of death receptors. (**A**) Protein expression analysis of the heat shock proteins HSP27 and HSP90, the tumor necrosis factor (TNF) receptor TNF-R1 and TNF-R2, TRAIL-R2, and the adaptor proteins FADD, TRADD, and RIP after treatment with 0.75 µM and 1.5 µM SK119 for 24 h in WM9 and MUG-Mel2 cells. Untreated control cells (ctrl) served as reference value and β-actin was used as loading control. One representative blot out of three is shown; (**B**) Densiometric quantifications are presented (mean ± SD; *n* = 3). All full-length blots are presented in the Appendix A.

**Figure 6 ijms-23-05684-f006:**
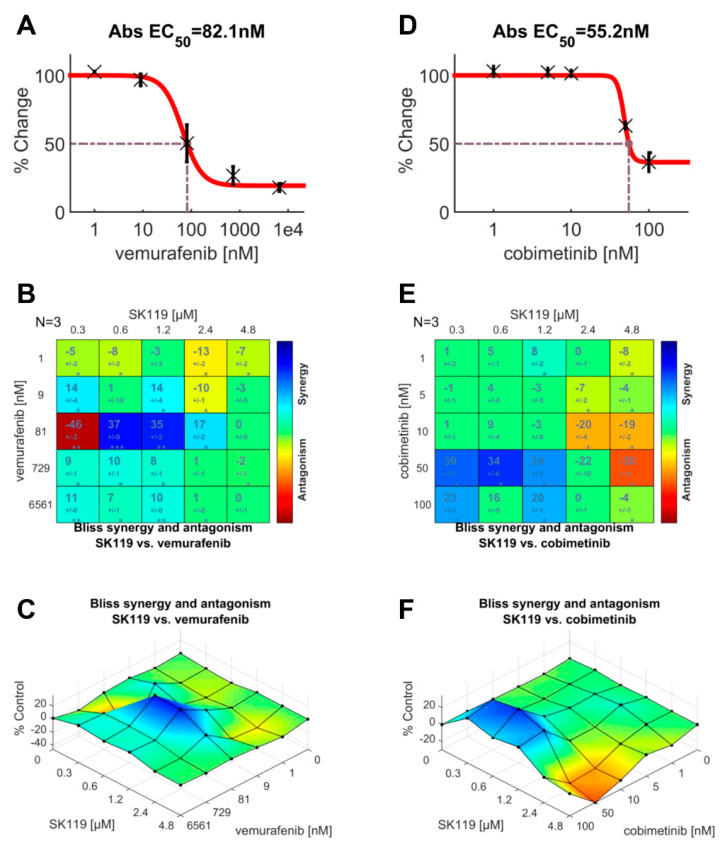
Treatment of WM9 and MUG-Mel2 cells with vemurafenib or cobimetinib and SK119. (**A–C**) treatment of WM9 cells with vemurafenib (**A**) or vemurafenib and SK119 in varying concentrations and analyzed using the Bliss model and the Combenefit software tool (**B**,**C**); (**D**–**F**) treatment of MUG-Mel2 cells with cobimetinib (**D**) or cobimetinib and SK119 in varying concentrations and analyzed using the Bliss model and the Combenefit software tool (**E**,**F**) (*n* = 3, incubation time: 72 h).

**Table 1 ijms-23-05684-t001:** NMR data of SK119, measured in CDCl_3_ and benzene-d_6_. *, **, *** Peaks with the same symbol may be interchanged within column. For numbering, see Figure 1.

	Solvent CDCl_3_		Solvent C_6_D_6_	
No.	^13^C	^1^H	^13^C	^1^H
1	186.8	-	187.3	-
2	143.9?	-	144.1	-
3	143.9	-	144.7	-
4	186.2	-	187.0	-
5	158.3 **	OH: 12.72 s	159.4	OH: 13.10 s
6	129.6 ***	7.20 s	130.1	6.70 d, *J* = 9.1 Hz
7	129.2 ***	7.20 s	129.8	6.69 d, *J* = 9.1 Hz
8	158.6 **	OH: 12.72 s	159.6	OH: 13.03 s
9	112.1 *	-	112.8	-
10	111.80 *	-	112.52 *	-
11	120.6	6.58 d, *J* = 15.6 Hz	121.8	6.76 d, *J* = 15.3 Hz
12	139.2	7.41 dd, *J* = 15.6, 11.3 Hz	139.9	7.65 dd, *J* = 15.3, 11.1 Hz
13	126.7	6.04 d, *J* = 11.2 Hz	127.8	6.03 d, *J* = 11.4 Hz
14	143.5	-	143.4	-
15	26.5	1.63 s	26.6	1.58 s
16	19.0	1.60 s	19.1	1.60 s
1′	182.5	-	182.79 **	-
2′	151.4	-	151.6	-
3′	135.7	7.23 d, *J* = 1.1 Hz	136.3	7.26 d, *J* = 0.9 Hz
4′	182.7	-	182.72 **	-
5′	162.5	OH: 12.47 s	163.6	OH: 12.85 s
6′	131.2	7.19 d, *J* = 9.6 Hz	131.8	6.59 d, *J* = 9.7 Hz
7′	131.0	7.16 d, *J* = 9.6 Hz	131.2	6.53 d, *J* = 9.7 Hz
8′	163.1	OH: 12.41 s	164.2	OH: 12.71 s
9′	111.77 *	-	112.2	-
10′	111.6	-	112.48 *	-
11′	38.8	4.67 tm, *J* = 7.6 Hz	39.6	4.72 tm, *J* = 7.0 Hz
12′	29.7	2.73–2.87 m, 2H	30.6	2.62–2.79 m, 2H
13′	120.3	5.08 tm, *J* = 6.8 Hz	121.5	5.14 tm, *J* = 7.0 Hz
14′	135.2	-	135.2	-
15′	25.7	1.89 s	26.1	1.53 s
16′	18.0	1.85 s	18.3	1.51 s

**Table 2 ijms-23-05684-t002:** IC_50_ values (µM) of SK119 after 72 h treatment compared to the most active, formerly isolated derivative DMAS; (mean ± SD, *n* = 6). x-fold ∆: x-fold difference of the IC_50_ values = IC_50 DMAS_/IC_50 SK119_. IC_50_ values were determined using the four-parameter logistic curve.

Cell Line	Sbcl2	WM9	WM164	MUG-Mel2	FS-1	HEK-293
SK119	0.3 ± 0.01	0.6 ± 0.01	1.7 ± 0.3	0.7 ± 0.01	0.5 ± 0.01	2.2 ± 0.4
DMAS	1.1 ± 0.1	2.7 ± 0.3	8.3 ± 0.3	7.2 ± 0.5	1.8 ± 0.3	9.5 ± 1.4
x-fold ∆	3.7	4.5	4.9	10.3	3.7	4.3

## Data Availability

Not applicable.

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
