# Peer review of "SK119, a Novel Shikonin Derivative, Leads to Apoptosis in Melanoma Cell Lines and Exhibits Synergistic Effects with Vemurafenib and Cobimetinib"

_ijms, 2022, doi:10.3390/ijms23105684_

Round 1

Reviewer 1 Report

The authors demonstrated synergistic effects of two drugs. 

They used several types of cell lines to strengthen their prediction, and I think authors performed appropriate methods of experiments. They carried out not only cell viability assay but also analyzed apoptotic pathway including caspase activities. 

Using software for assessing synergistic effect of drugs is beneficial reference for future research. 

Taken together, I think this paper is acceptable for IJMS.

Author Response

The authors sincerely thank the reviewer for his/her efforts and the time he/she spent on this.

We are very pleased about the positive assessment of our study!

Reviewer 2 Report

In cancer prevention and therapeutic the natural products and its derivatives plays very important role and indeed exploration of novel natural remedies will greatly helpful. The authors have nicely designed the experimental study the role of novel shikonin derivative SKR119 by including diverse cell line of melanoma and controls. Since the SK119 shows the strong apoptosis inducer and this study have done only in in-vitro stage, there are some concerns need to address:

  1. To protect the normal cellular environment to prevent desired side effect with minimal cytotoxicity, the pharmacokinetics and pharmacodynamics of SK119 needs to be validate in rigorous manner.
  2. The authors reported that the compound SK119 is the strong inducer of apoptosis and the Cytotoxic effect of SK119 prevails in non-tumorigenic cell line FS1 and HEK293, How the author assures the prevention of cytotoxicity in in-vivo study models.
  3. In order to establish the therapeutic efficacy, have the authors studied the effect of SK119 on the skin pigmentation prospectus melanocytes.
  4. In the title of the manuscript need to fix Mel-Anoma by replacing melanoma and Vemu-Rafenib by vemurafenib

Author Response

In cancer prevention and therapeutic the natural products and its derivatives plays very important role and indeed exploration of novel natural remedies will greatly helpful. The authors have nicely designed the experimental study the role of novel shikonin derivative SKR119 by including diverse cell line of melanoma and controls. Since the SK119 shows the strong apoptosis inducer and this study have done only in in-vitro stage, there are some concerns need to address:

1. To protect the normal cellular environment to prevent desired side effect with minimal cytotoxicity, the pharmacokinetics and pharmacodynamics of SK119 needs to be validate in rigorous manner.

We thank the reviewer for her/his constructive suggestions. We agree that this needs to be address in detail. For further studies, we plan to perform comprehensive in vitro and in vivo studies to elucidate the potential of SK119 in more detail including its cytotoxicity profile.

2. The authors reported that the compound SK119 is the strong inducer of apoptosis and the Cytotoxic effect of SK119 prevails in non-tumorigenic cell line FS1 and HEK293. How the author assures the prevention of cytotoxicity in in-vivo study models?

Cytotoxicity of chemotherapeutics towards healthy cells is a known problem in cancer therapy and can lead to undesired side effects. We cannot fully exclude that SK119 will also cause undesired side effects in vivo when applied in its current form. However, there are ways which might help to overcome or reduce these negative effects. One might be to use smart-loaded targeted nanoparticles. It has been reported that blood vessels of tumors are leaky allowing nanoparticles to penetrate specifically into the tumor tissue. In addition, lymphatic drainage in tumors is poor retaining the accumulated nanoparticles and allowing the drug to be released. Moreover, shikonin-loaded nanoparticles improved the anti-tumor effects of shikonin in glioma cells in vitro and the particles accumulated in the brain of rats. For melanoma, it has been demonstrated that self-assembled nanomicelles of clotrimazole improve drug delivery and apoptosis and, at the same time, inhibit tumor progression. Therefore, we assume that this might be a promising way for further research. However, development, characterization as well as in vitro and in vivo testing of SK119 and such nanoparticles goes beyond the scope of the current work.

3. In order to establish the therapeutic efficacy, have the authors studied the effect of SK119 on the skin pigmentation prospectus melanocytes?

Not yet. However, during our further studies, we will also include isolation of melanocytes from healthy patients and investigate the effect of SK119 in these freshly isolated cells.

4. In the title of the manuscript need to fix Mel-Anoma by replacing melanoma and Vemu-Rafenib by vemurafenib

Thank you very much for pointing this out. This error must have occurred during the editing process by the publisher. Has been corrected.